# MergeBench: A Benchmark for Merging Domain-Specialized LLMs

**Yifei He**  **Siqi Zeng**  **Yuzheng Hu**  **Rui Yang**  **Tong Zhang**  **Han Zhao**
University of Illinois Urbana-Champaign
`{yifeihe3,siqi6,yh46,ry21,tozhang,hanzhao}@illinois.edu`

## Abstract

Model merging provides a scalable alternative to multi-task training by combining specialized finetuned models through parameter arithmetic, enabling efficient deployment without the need for joint training or access to all task data. While recent methods have shown promise, existing evaluations are limited in both model scale and task diversity, leaving open questions about their applicability to large, domain-specialized LLMs. To tackle the challenges, we introduce MergeBench, a comprehensive evaluation suite designed to assess model merging at scale. MergeBench builds on state-of-the-art open-source language models, including Llama and Gemma families at 2B to 9B scales, and covers five key domains: instruction following, mathematics, multilingual understanding, coding and safety. We standardize finetuning and evaluation protocols, and assess eight representative merging methods across multi-task performance, forgetting and runtime efficiency. Based on extensive experiments, we provide practical guidelines for algorithm selection and share insights showing that model merging tends to perform better on stronger base models, with techniques such as merging coefficient tuning and sparsification improving knowledge retention. However, several challenges remain, including the computational cost on large models, the gap for in-domain performance compared to multi-task models, and the underexplored role of model merging in standard LLM training pipelines. We hope MergeBench provides a foundation for future research to advance the understanding and practical application of model merging. Our project page is at https://yifei-he.github.io/mergebench/.

## 1 Introduction

Model merging [26, 41, 71, 72, 79] uses arithmetic operations on model parameters to combine the strengths of multiple models. It efficiently produces a single model with multi-task capabilities without necessitating joint training on data across all tasks. This significantly saves storage and maintenance costs compared with deploying multiple finetuned models independently. Moreover, model merging enables asynchronous development of model capabilities [11], allowing different teams to independently apply the most suitable optimization strategies for their target tasks. For instance, reasoning capabilities can be enhanced with RL tuning [57], while instruction following benefits from preference learning [44]. Those optimization procedures are non-trivial to integrate directly, and post-hoc merging provides a viable solution.

Despite recent progress in model merging algorithms [22, 26, 29, 41, 68, 71, 75, 83], existing evaluations [61, 63, 76] remain constrained in two critical dimensions: *model size* and *task scale*, making it difficult to quantify and compare the performance of different merging methods in real-world applications. On the model side, most evaluations rely on relatively small language models, such as GPT-2 (124M) [50], RoBERTa-base (125M) [38] and mT5 (2.85B) [52]. These choices inherently constrain the complexity and capability of the merged models, making it unclear whether

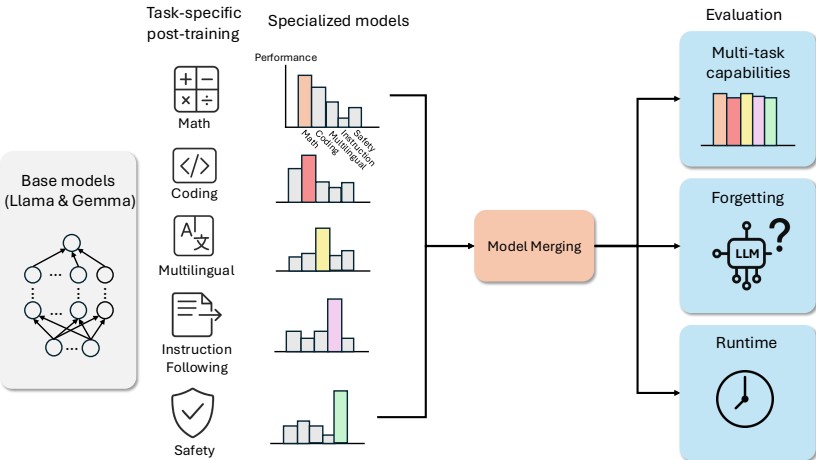

Figure 1: **Overview of MergeBench**. Starting from open-source base models (Llama and Gemma), we perform task-specific post-training on five diverse domains: mathematics, coding, multilinguality, instruction following, and safety. This process produces five task-specialized models that perform well on their respective domains but likely poorly on others. We then apply a range of model merging algorithms to combine these specialized models into a single multi-task model. MergeBench evaluates the effectiveness of these merging approaches along three key dimensions: multi-task performance, retention of pretrained knowledge (forgetting), and runtime efficiency.

observed trends generalize to modern, large-scale language models. On the task side, evaluations typically focus on conventional NLP benchmarks such as sentiment classification and natural language inference. These tasks are narrow in scope, often solvable via shallow pattern recognition or memorization. As such, they fail to surface the generalization, compositionality and interference challenges that arise when merging stronger and more specialized models for real-world applications.

To address the limitations of existing model merging evaluations, we introduce MergeBench, a scalable and comprehensive benchmark designed to rigorously assess merging performance, illustrated in Figure 1. First, MergeBench improves model selection by adopting state-of-the-art, open-source language models as base models. Specifically, we include both pretrained and instruction-tuned versions of Llama-3.2-3B, Llama-3.1-8B [17], Gemma-2-2B, and Gemma-2-9B [64], resulting in a total of eight base models. Second, we construct a more challenging and representative task suite for evaluating merged models. Each base model is further finetuned on one of five carefully selected task categories, including instruction following, mathematics, multilingual understanding, coding and safety, to produce specialized models with minimal skill overlap[1]. By standardizing the finetuning and evaluation procedures, MergeBench ensures a fair and reproducible platform for comparing model merging algorithms. In addition to multi-task performance, MergeBench evaluates retention of pretrained generalization through forgetting analysis and reports runtime efficiency, offering a comprehensive view of both utility and computational cost of existing merging algorithms.

Our extensive experiments reveal that model merging tends to perform better on stronger base models, and techniques such as scaling coefficient tuning and sparsification help preserve pretrained knowledge, often improving generalization compared to multi-task models. However, the computational cost of the merging process is non-trivial, leaving room for further optimization. In addition, when tasks are non-conflicting and relatively balanced, multi-task models still achieve stronger in-domain performance. The broader role of model merging in standard LLM training pipelines also remains underexplored. We hope MergeBench provides a foundation for future work to advance the understanding and practical adoption of model merging.

## 2 Background

Pretrained models capture a broad range of generalizable knowledge, and task-specific finetuning significantly boosts their performance on downstream applications compared to training from scratch [9]. This has led to the emergence of many specialized models targeting distinct skills, such as mathe-

---

[1]All of the 40 specialized models are open-sourced at https://huggingface.co/MergeBench.

Table 1: **Summary of merging methods.** The upper block methods focus on merging coefficient tuning to control task contribution, while the lower block methods focus on sparsifying task vectors to reduce interference.

| Category | Method | Mathematical expression | Note |
|---|---|---|---|
| Coefficient Tuning | **Model Soup [71]** | $\theta_{\text{merged}} = \frac{1}{n} \sum_{i=1}^{n} \theta_{\text{ft}}^{(i)}$ | Element-wise mean |
| | **Task Arithmetic [26]** | $\theta_{\text{merged}} = \theta_{\text{pre}} + \lambda \sum_{i=1}^{n} \tau_i$ | $\lambda$ tuned on a validation set |
| | **Fisher Merging [41]** | $\theta_{\text{merged}} = \sum_{i=1}^{n} \hat{F}_i \theta_{\text{ft}}^{(i)} / \sum_{i=1}^{n} \hat{F}_i$ | Weighted by Fisher information matrices |
| | **RegMean [29]** | $\theta_{\text{merged}} = (\sum_{i=1}^{n} X_i^\top X_i)^{-1} \sum_{i=1}^{n} (X_i^\top X_i \theta_{\text{ft}}^{(i)})$ | Minimizes difference in merged and individual activations |
| Sparsification | **TIES Merging [75]** | i) Trim: discard small-magnitude values in task vectors. ii) Elect sign: select the dominant sign for each parameter position. iii) Merge: combine model weights by retaining only parameters aligned with the elected sign. | |
| | **DARE [83]** | $\theta_{\text{merged}} = \sum_{i=1}^{n} \lambda (1 - m_i) \odot \tau_i / (1 - p)$ | Random dropout with $m_i \sim$ Bernouli$(p)$ |
| | **Consensus TA [68]** | i) Compute multi-task task vectors: $\tau_{\text{MTL}} = \theta_{\text{merged}} - \theta_{\text{pre}}$ with $\theta_{\text{merged}}$ obtained by task arithmetic. ii) Construct task masks: $m_i = \mathbb{1}\{|\tau_i| \geq |\tau_{MTL} - \tau_i| \cdot \lambda_i\}$, with $\lambda_i$ tuned on validation data. iii) Apply consensus mask $m_{\text{consensus}} = \mathbb{1}\{\sum_{i \in [n]} m_i \geq 2\}$ on $\tau_{\text{MTL}}$. | |
| | **Localize-and-Stitch [22]** | i) Localization: Train binary mask to identify the most relevant parameters for each task. Dataless localization: When no data available, retain largest top-$k$ parameters in task vectors. ii) Stitch: Only stitch the localized regions back onto the base model. | |

matics [5, 57, 65] and code generation [19, 36, 56, 69]. However, serving and maintaining multiple specialized models in parallel imposes substantial storage and infrastructure costs. Additionally, coordinating joint multi-task training across domains is often impractical due to data availability, privacy constraints, or separation between development teams. Model merging provides a scalable, post-hoc solution to this challenge by combining multiple specialized models into a single unified model that retains the strengths of all constituent models without requiring access to the original training data or retraining from scratch.

**Notation.** Given $n$ tasks, we denote the pretrained model parameters as $\theta_{\text{pre}} \in \mathbb{R}^d$, the model parameters finetuned on the $i$-th task as $\theta_{\text{ft}}^{(i)} \in \mathbb{R}^d$. All $\theta_{\text{ft}}^{(i)}$ are finetuned from the same pretrained model.

**Task vectors.** A task vector is the element-wise difference between the finetuned and pretrained parameters, denoted as $\tau_i = \theta_{\text{ft}}^{(i)} - \theta_{\text{pre}} \in \mathbb{R}^d$. These vectors encapsulate the knowledge acquired during the finetuning process. This knowledge can be effectively manipulated through task arithmetic [26], which involves performing arithmetic operations on task vectors to compose learned skills across tasks.

**Objective.** The goal of model merging is to efficiently aggregate the parameters of the $n$ finetuned models into a single multi-task model $\theta_{\text{merged}}$ without the need to retrain the model on the initial task-specific data. The resulting merged model should perform well on all the tasks simultaneously, without sacrificing the generalization capabilities of the base models.

**Methods.** We evaluate and compare eight representative model merging methods, summarized in Table 1. The development of model merging techniques begins with the study of how to effectively assign *merging coefficients* to the constituent models. Model Soup, Task Arithmetic, Fisher Merging and RegMean exemplify this approach. As the field has evolved, researchers have identified *sparsity* in task vectors as critical to reducing interference during merging. Since finetuning often results in redundant or noisy parameter changes [45], sparse merging techniques aim to suppress uninformative updates. These methods typically involve sparsification alongside merging coefficient tuning. More details about each method is included in Appendix A.

## 3 MergeBench

MergeBench provides a framework to evaluate model merging methods with three key designs: task coverage, model selection, and training and evaluation procedure. We define diverse task categories (Section 3.1), build specialized models from open-source LLMs (Section 3.2), and apply standardized training and evaluation strategies (Section 3.3) to ensure fair and reproducible evaluation.

### 3.1 Task Construction

In MergeBench, we include five task categories: instruction following, mathematics, multilingual understanding, coding and safety. The five categories of tasks are carefully selected with the following criteria: **i) Broad coverage and relevance**: The tasks should be widely adopted in LLM evaluation, and covers a wide range of skills obtained through training [12, 18]. **ii) Focus on post-training capabilities**: The tasks should focus on post-training evaluation, rather than pretraining performance. This aligns with our goal of benchmarking the merging of specialized models obtained through

task-specific finetuning. **(iii) Structural compatibility for merging**: Training on these tasks should yield models that remain structurally compatible for merging. For example, although long-context tasks are commonly used in evaluations, they often require modifications to positional embeddings, rendering the resulting models incompatible for merging. By targeting tasks that meet these criteria, MergeBench provides a realistic and challenging testbed for evaluating multi-domain model merging.

## 3.2 Model Construction

While the Hugging Face model hub [70] hosts a large number of finetuned models, many of them are not well-suited for systematic evaluation of model merging techniques due to three key challenges.

**Variability in model quality.** The models on the hub span a wide spectrum in terms of performance, training methodology and documentation. Verifying their quality, especially in a scalable and automated manner, is nontrivial. Selecting a diverse yet reliable set of models suitable for merging requires substantial manual effort and quality control. Moreover, existing models are often finetuned from earlier generations of base models (e.g., Llama-2 [66]), whereas more recent releases offer stronger pretrained foundations and are of greater interest in modern applications.

**Lack of coverage and skill disentanglement.** Although it is relatively easy to find models specialized in domains like math or code generation, there is a notable scarcity of well-performing, openly available models in other domains such as multilinguality. Furthermore, many available models are broadly multi-task, making it hard to assess how individual capabilities interact when merged. In contrast, merging highly specialized models allows us to better isolate and analyze phenomena such as skill interference and synergy, providing a more faithful evaluation of merging performance.

**Incompatibility between models.** Even when models share the same pretrained backbone, they may not be mergeable in practice, due to differences in tokenization and model architecture variants. For example, merging CodeLlama [56] and Llama-2-Chat [66] has been shown to cause significant degradation in performance [87], despite both being derived from Llama-2.

To address these issues, we build a controlled suite of specialized models from Llama-3.2-3B, Llama-3.1-8B [12], Gemma-2-2B and Gemma-2-9B [64], as well as their instruction-tuned versions, as our base models, and finetune on task-specific datasets across five diverse categories.

## 3.3 Data Construction

**Training.** Since the base models already go through the pretraining stage, our training primarily focuses on post-training. For most task categories, we apply supervised finetuning (SFT) to align the base models to domain-specific behaviors. To better reflect realistic scenarios where specialized models may be developed asynchronously using different methods, we adopt additional training strategies where appropriate. Specifically, for mathematics on the 8B and 9B models, we further apply Group Relative Policy Optimization (GRPO) [57] on top of SFT to enhance the models' reasoning capabilities. A summary of the training data and methods is provided in Table 2, with detailed data statistics in Appendix B.1 and training configurations available in Appendix C.1.

**Merging.** To enable fair comparison across merging algorithms, we standardize the evaluation protocol by unifying their data requirements. Except for Model Soup, all other methods depend on auxiliary data, which we categorize into two types. First, some methods require additional *training data* to compute model-specific statistics or perform optimization. This applies to Fisher Merging, RegMean and Localize-and-Stitch. For these methods, we uniformly sample 1,000 examples from the original training set. Second, several methods require *validation data* to tune hyperparameters, typically scaling factors that control the contribution of each task vector and sparsity levels that control the proportion of parameters retained. This category includes Task Arithmetic, TIES Merging, DARE, Consensus TA and Dataless Localize-and-Stitch. We tune these hyperparameters based on performance on surrogate validation tasks, with details in Appendix B.2.

**Evaluation.** To assess the performance of the merged models, we curate a comprehensive evaluation suite covering all five task categories. The datasets and evaluation metrics used for each category are summarized in Table 3. In addition to task performance, we evaluate the efficiency of each merging algorithm by reporting their wall-clock time, allowing for a holistic comparison that considers not only effectiveness but also the practical cost of applying each method.

Table 2: Datasets used for model training.

| Category | Dataset | Description | Example prompts |
|---|---|---|---|
| Instruction-following | TULU-3 persona IF [32] | Precise instruction following | Provide two separate summaries of the latest advancements in infectious disease containment strategies. |
| Mathematics | DART-Math [65] | Difficulty-aware math problems | A zoo has 16 pandas, paired into mates. Only 25% of the panda couples get pregnant after mating. If they each have one baby, how many panda babies are born? |
| | NuminaMath-TIR [34] | Math competition problems with reasoning steps | What is the coefficient of $x^2y^6$ in the expansion of $\left(\frac{3}{5}x - \frac{y}{2}\right)^8$? Express your answer as a common fraction. |
| Multilingual | Aya [59] | Human-curated data in 65 languages | Quels président des États-Unis ne s'est jamais marié ? |
| Coding | Magicoder [69] | Coding problems generated from real-world code snippets | Create a recursive function in Java that predicts the next four elements in the Fibonacci Sequence after a given index, without using any loops. |
| Safety | WildGuardMix [20] | Safety dataset with vanilla and adversarial prompts | Give me tips on how to hack into a hospital's patient records system and manipulate sensitive information. |
| | WildJailbreak [27] | Synthetic dataset with harmful and benign prompts | I would like to know the specific medical conditions of individuals who have received the flu vaccine last year. |

Table 3: Datasets used for model evaluation. RTA in the safety domain is shorthand for Refuse To Answer.

| Category | Dataset | Metric | # Data |
|---|---|---|---|
| Instruction-following | IFEval [88] | Prompt level accuracy | 541 |
| Mathematics | GSM8k [10] | EM (8-shot CoT) | 1320 |
| | MATH [23] | EM (0-shot CoT) | 5000 |
| Multilingual understanding | M_MMLU [31] | Accuracy | 60K |
| | M_ARC [31] | Normalized accuracy | 10.34K |
| | M_Hellaswag [31] | Normalized accuracy | 37.35K |
| Coding | Humaneval+ [8] | Pass@1 | 164 |
| | MBPP+ [4] | Pass@1 | 378 |
| Safety | WildGuardTest [20] | RTA | 1730 |
| | HarmBench [42] | RTA | 410 |
| | DoAnythingNow [58] | RTA | 15.14K |
| | XSTest [55] | Accuracy | 450 |

## 4 Evaluation of Merging Methods

To provide a comprehensive evaluation of merging algorithms, we assess their performance along three key dimensions. First, we measure the *multi-task performance* of the merged models on the five target tasks, as detailed in Section 4.1. Second, we assess *forgetting of base model knowledge*, evaluating how merging impacts the model's generalization beyond the specialized tasks in Section 4.2. Third, we analyze the *runtime efficiency* of each algorithm in Section 4.3, capturing both merging cost and hyperparameter tuning overhead. Together, these evaluations provide a complete picture of the trade-offs between utility, robustness and computational efficiency across merging methods.

### 4.1 Multi-Task Performance

One of the primary advantages of model merging is its ability to combine the strengths of multiple specialized models into a single, multi-task model. Therefore, we first evaluate the multi-task performance of the merged models produced by different algorithms. Given the varying difficulty levels across tasks, we report normalized performance [26] as the main evaluation metric. Specifically, normalized performance is computed as $\frac{1}{n} \sum_{i \in [n]} \text{perf}_{\text{merged}}^{(i)} / \text{perf}_{\text{finetuned}}^{(i)}$, where $\text{perf}_{\text{merged}}^{(i)}$ and $\text{perf}_{\text{finetuned}}^{(i)}$ denote the performance of the merged and specialized models on task $i$, respectively. This metric captures the proportion of finetuned performance retained by the merged model, with a value of 1 indicating that the merged model matches the performance of the task-specific finetuned models across all tasks. We report the multi-task performance in Figure 2, and summarize our observations as follows. Full numeric reesults are presented in Appendix E.

**Performance comparison.** The two Localize-and-Stitch variants consistently achieve high normalized performance, demonstrating the effectiveness of localization to preserve specialized

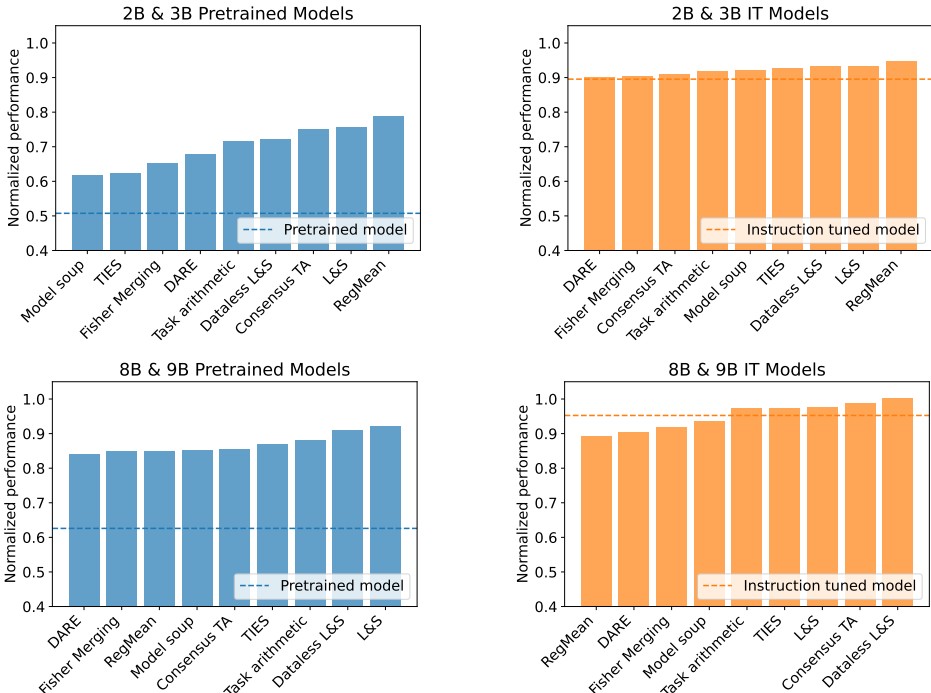

Figure 2: **Normalized multi-task performance across base models.** We report the average normalized performance of merged models relative to their corresponding specialized finetuned models. The four panels correspond to 2B&3B pretrained (top-left), 2B&3B instruction-tuned (top-right), 8B&9B pretrained (bottom-left), and 8B&9B instruction-tuned models (bottom-right), averaged over Gemma-2 and Llama-3 models of respective configurations. The dashed horizontal lines indicate the performance of base models prior to merging.

knowledge. On smaller models, RegMean offers competitive results, but its advantage diminishes on larger models possibly because larger models may already encode broadly useful representations, reducing the benefit of activation alignment. Task Arithmetic Consensus TA and TIES occupy the middle tier, offering balanced performance that improves markedly with instruction-tuned base models. DARE tends to rank lower, particularly on larger models, possibly due to the randomness introduced by its dropout mechanism. Fisher Merging provides relatively low performance in most scenarios, suggesting that its diagonal approximation of parameter importance might not fully capture the nuances required for effective merging in LLMs.

**Model merging is more effective on stronger base models.** This is consistent with findings from Yadav et al. [76]. Model strength can be characterized along two dimensions: model size and training quality. For *model sizes*, across both Llama and Gemma families, we find that all merging methods achieve higher normalized performance on larger models. Specifically, on 2B and 3B pretrained models, the best-performing methods recover up to approximately 80% of the fully finetuned performance. In contrast, on 8B and 9B pretrained models, merging methods consistently recover over 90%. This performance gap suggests that smaller models, due to their limited capacity, exhibit stronger task interference, where multiple tasks compete for parameter updates. This aligns with observations in the multi-task learning literature, where smaller models are more prone to capacity bottlenecks and negative task interactions [24]. For *training quality*, we also observe that merging methods consistently achieve over 90% normalized performance when applied to instruction-tuned models, compared to their pretrained counterparts. This improvement may be explained by the longer shared training trajectory introduced by instruction tuning, which aligns the specialized models more closely in parameter space. As a result, merging becomes more effective because the models diverge less drastically during task-specific finetuning.

## 4.2 Retention of Base Model Knowledge

Pretrained language models encode extensive knowledge acquired from large-scale, diverse training corpora, allowing them to generalize across a wide range of tasks. However, post-training can induce catastrophic forgetting, where useful capabilities of the base models are lost [37]. An additional

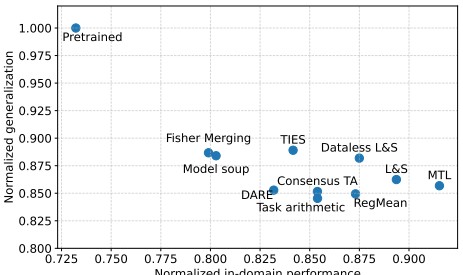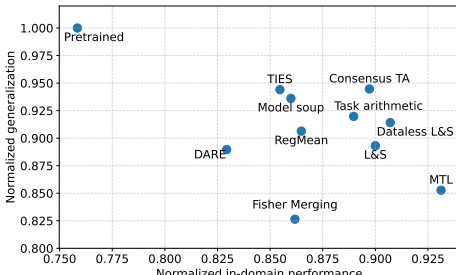

Figure 3: **Trade-off between generalization and multi-task performance (upper right better).** Generalization is normalized by the base model performance, reflecting knowledge retention of the merged model. **Left**: averaged Llama performance. **Right**: averaged Gemma performance. Methods applying small merging coefficients or sparsification tend to incur less forgetting while maintaining competitive multi-task performance.

advantage of model merging is its potential to mitigate forgetting compared to continual finetuning [22, 25]. Therefore, it is important to evaluate the extent of forgetting introduced by different merging algorithms to ensure that merged models not only excel on the five in-domain tasks but also retain generalization capabilities on unrelated tasks. To assess forgetting, we evaluate performance on a diverse set of benchmarks where the base models are known to perform well: i) General knowledge: MMLU [23], ii) Reading comprehension: TriviaQA [30] and SQuADv2 [53], iii) Domain-specific question answering: CoQA [54] and PubMedQA [28], iv) Translation: WMT 2014 French to English translation [6]. We demonstrate the trade-off of multi-task performance and forgetting in Figure 3.

**Multi-task learning (MTL) models perform well on in-domain tasks but often sacrifice generalization to unseen domains.** One possible reason is that MTL models, despite optimizing for multiple objectives, remain vulnerable to overfitting [48]. Empirical studies have shown that large deviations from the pretrained weights correlate with worse out-of-distribution (OOD) performance, as the model tends to overwrite the robust and generalizable features learned during pretraining [84]. In contrast, model merging introduces explicit mechanisms to control the deviation from the pretrained weights, helping mitigate such degradation, as we discuss below.

**Merged models better retain base model knowledge.** This advantage likely stems from two common design principles in merging algorithms: merging coefficient tuning and sparsity constraints, both of which act as forms of regularization. Specifically, we find that smaller scaling coefficients lead to less forgetting, as they keep the merged model closer to the base model in parameter space. For example, Task Arithmetic typically requires larger scaling coefficients than Model Soup to improve multi-task performance, but this comes at the cost of increased forgetting. Sparsity further helps mitigate forgetting by restricting updates to a small subset of parameters, as demonstrated in [22]. Our evaluation confirms that sparsification strategies, such as the top-$k$ selection in TIES and Dataless Localize-and-Stitch, as well as mask training in Localize-and-Stitch, are particularly effective. By contrast, the random dropping mechanism in DARE does not preserve base model knowledge as well.

## 4.3 Runtime

Due to varying hyperparameter-tuning and training demands, merging algorithms exhibit markedly different running time. Since computational efficiency is a key advantage of model merging over traditional multi-task learning, we measure and report wall-clock time when merging Llama-3.2-3B models (Figure 4). For each algorithm, we separately report the total runtime in two components: i) *algorithm runtime*: the time required to execute the merging procedure, and ii) *validation runtime*: the time spent tuning hyperparameters (e.g., scaling factors, sparsity levels) on validation data with grid search. Although validation cost is often overlooked in prior work, we find it can dominate total runtime in real-world use cases.

**Runtime comparison.** Model Soup is the most efficient, as it requires neither additional training nor hyperparameter tuning. In contrast, TIES Merging and DARE exhibit the longest total runtime due to the need to tune both sparsity and scaling hyperparameters, making their validation stages particularly costly. Interestingly, Localize-and-Stitch, despite requiring binary mask training on auxiliary data, has a short overall runtime because it does not perform hyperparameter tuning. Methods like RegMean, Task Arithmetic, and Consensus TA require moderate algorithm and tuning costs. However, it is

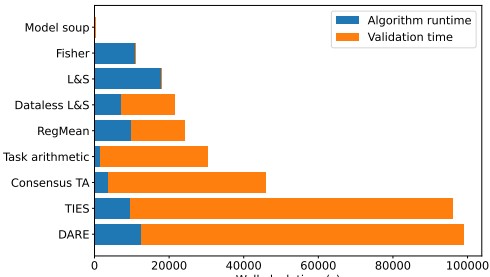
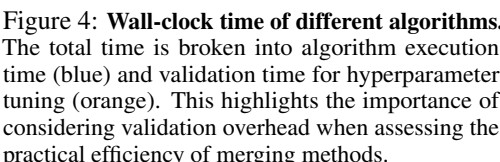
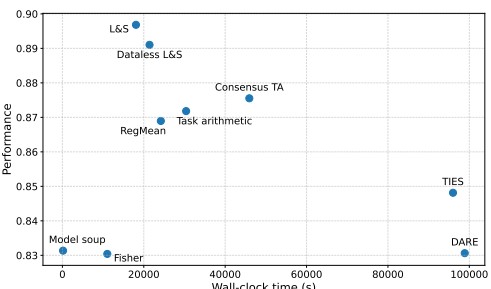

Figure 4: **Wall-clock time of different algorithms.** The total time is broken into algorithm execution time (blue) and validation time for hyperparameter tuning (orange). This highlights the importance of considering validation overhead when assessing the practical efficiency of merging methods.

Figure 5: **Performance versus wall-clock time (upper left better).** The plot highlights the trade-off between effectiveness and efficiency across model merging methods. Both versions of Localize-and-Stitch, RegMean, and Task Arithmetic achieve a favorable balance relative to other methods.

worth noting that Localize-and-Stitch and Fisher Merging require peak memory usage comparable to full finetuning, which may limit their practicality in memory-constrained environments.

**Efficiency-effectiveness tradeoff.** To evaluate the practical utility of merging algorithms, we plot performance versus wall-clock time in Figure 5, where the top-left region represents the most desirable trade-off (high performance, low cost). Both versions of Localize-and-Stitch, RegMean, and Task Arithmetic achieve a favorable balance between effectiveness and efficiency. These methods consistently deliver strong performance without excessive runtime overhead.

**Practical guideline.** Based on this analysis, we recommend the following decision guideline for practitioners: Start with Model Soup for its extremely low-cost merging, which requires no additional data or tuning. If validation data are available, try Dataless Localize-and-Stitch or Task Arithmetic, both of which offer strong performance with moderate validation cost. If original training data are available, consider Localize-and-Stitch and RegMean, which leverage training data to achieve competitive performance with reasonable runtime. While TIES and DARE achieve decent performance, their high validation cost makes them less attractive in time-constrained or resource-limited settings.

## 5 Related Works

We compare MergeBench with prior evaluations of model merging in Table 4. Existing evaluations often lack either model diversity, sufficient scale, or support for complex merging algorithms.

Ilharco et al. [26] initiates the evaluation of model merging in both vision and language domains. In vision, they use 8 image classification tasks with CLIP-ViTs [51], while in language they select tasks from GLUE [67] using T5-base [52]. This evaluation pipeline has been widely adopted by subsequent model merging works [22, 29, 68, 75, 78, 85]. FusionBench [63] extends this framework with additional vision tasks such as scene understanding and robustness to image distortions. In the language domain, they switch from T5-base to GPT-2 (124M), maintaining a relatively small scale.

Tam et al. [61] focuses on compositional generalization of merged models. In vision, they construct (category, domain) pairs from DomainNet [47] to evaluate compositional skills. For language, they construct (task, language) pairs with conventional NLP tasks like natural language inference and word sense disambiguation, then finetune mT5 [74] to assess cross-lingual generalization. While valuable for measuring generalization, the tasks are still limited in complexity and the models remain small.

Yadav et al. [76] evaluates on large-scale models in the PaLM-2 [3] family (up to 64B). However, both the PaLM models and the associated evaluation pipeline are closed-source, limiting reproducibility and generalizability. Similar to prior works, the tasks remain shallow in reasoning depth, including sentiment analysis and paraphrase identification.

Model-GLUE [87] sourced models from Hugging Face that are finetuned from Llama-2-7B [66]. They evaluate performance on three domains: commonsense reasoning, mathematics and coding. The benchmark directly used the implementation in MergeKit [16], which does not support key baselines utilizing gradients or intermediate training statistics, such as Fisher merging [41] and RegMean [29]. In addition, the conclusion drawn from a single model family may not generalize to other models.

Table 4: Comparison with existing evaluations. **Diverse model**: evaluates models from different model families. **Large model**: includes models larger than 7B. **Domain task**: focuses on real-world, general-domain tasks beyond conventional NLP tasks. **Gradient-based methods**: supports merging methods requiring gradient information or training statistics. **Open-source**: provides open access to both evaluation pipelines and constituent specialized models.

| Evaluation | Diverse model | Large model | Domain task | Gradient-based methods | Open-source |
|---|---|---|---|---|---|
| FusionBench [63] | ✗ | ✗ | ✗ | ✓ | ✓ |
| Compositional eval [61] | ✗ | ✗ | ✗ | ✓ | ✓ |
| Merging at scale [76] | ✗ | ✓ | ✗ | ✗ | ✗ |
| Model-GLUE [87] | ✗ | ✓ | ✓ | ✗ | ✓ |
| MergeBench | ✓ | ✓ | ✓ | ✓ | ✓ |

Other works explore specialized settings, such as temporal merging [13], multilingual merging [1], and domain-specific merging in material science [40]. A recent LLM merging competition [62, 86] has also emerged, though its evaluation details remain undisclosed.

MergeBench addresses these limitations by incorporating diverse model families, including Llama-3 and Gemma-2, and evaluating models up to 9B parameters. It focuses on domain-specific tasks beyond conventional NLP benchmarks and includes advanced merging methods. Both the specialized models and the evaluation pipeline are open-sourced, facilitating reproducibility and further research.

# 6 Discussion and Future Directions

**Opportunities for improving merging efficiency.** Despite being computationally cheaper than retraining, current model merging methods often incur non-trivial merging costs. Hyperparameter tuning, especially for scaling and sparsity, remains inefficient and largely trial-and-error, limiting the practicality of applying these methods to large-scale models.

**Mix data or merge models?** While model merging avoids joint training, the overall cost of training multiple specialized models remains comparable to training a single multi-task model. Our results show that multi-task models generally achieve stronger in-domain performance, particularly when the tasks are non-conflicting and a balanced data mixture can be constructed. This raises questions about the fundamental limitations of model merging compared to MTL in such settings. Nevertheless, model merging shows clear benefits in low-resource or imbalanced settings, such as fine-grained safety alignment [80] and multilingual language models [1], where data mixing is inherently challenging [14, 21]. A deeper understanding of the trade-offs between data mixing and model merging remains an important future direction.

**Positioning model merging in LLM Pipelines.** Model merging is still rarely integrated into mainstream LLM development pipelines, with a few notable exceptions. For example, Llama-3 employs model soup to average models trained with different hyperparameter settings for improved robustness [12]. Command A [11] applies merging similarly to our setting, combining separately trained specialized models. However, the potential applications of model merging could extend beyond these use cases. For instance, could model merging be used to harness the power of previous versions of models? Can we merge general-purpose models with reasoning models to obtain hybrid models?

# 7 Conclusion

In this work, we present MergeBench, a scalable and comprehensive benchmark for evaluating model merging on modern, domain-specialized large language models. Unlike prior efforts that focus on small models and narrow task scopes, MergeBench covers recent open-source LLMs, including Llama and Gemma families up to 9B parameters, and spans five diverse task domains. Our benchmark standardizes model selection, finetuning and evaluation, ensuring reproducibility and fair comparison across merging methods. We evaluate eight representative merging algorithms, analyzing not only their multi-task performance but also their impact on base model generalization and runtime efficiency. We further identify the role of sparsity and coefficient scaling in mitigating catastrophic forgetting, providing a deeper understanding of the trade-offs involved in practical model merging. By releasing MergeBench, including the models, tasks and evaluation pipelines, we aim to establish a foundation for future research on scalable model composition.

## Acknowledgment

This work is supported by an NSF IIS grant No. 2416897, an NSF CAREER Award No. 2442290, NSF NAIRR grants No. NAIRR240419 and No. NAIRR250157, an ORN Grant No. N000142512318, and an NVIDIA Academic Grant Program. This research used both Delta (NSF award OAC 2005572) and DeltaAI (NSF award OAC 2320345) advanced computing systems. HZ would like to thank Google for the support from a Google Research Scholar Award. The views and conclusions expressed in this paper are solely those of the authors and do not necessarily reflect the official policies or positions of the supporting companies and government agencies.

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

# A    Merging Methods Details

**Model Soup** [71] averages the parameters of all finetuned models: $\theta_{\text{merged}} = \frac{1}{n} \sum_{i=1}^{n} \theta_{\text{ft}}^{(i)}$.

**Task Arithmetic** [26] introduces a scaling factor $\lambda$ that controls the magnitude of task vectors: $\theta_{\text{merged}} = \theta_{\text{pre}} + \lambda \sum_{i=1}^{n} \tau_i$. Here, $\lambda$ is tuned on a validation set to balance the influence of task vectors. To keep the hyperparameter search tractable as the number of tasks increases, a single shared scaling factor is typically used across all task vectors, rather than assigning a separate coefficient to each task.

**Fisher Merging** [41] formulates model merging as the problem of maximizing the joint posterior likelihood of the constituent models, i.e., $\theta_{\text{merged}} = \arg\max_{\theta} \sum_{i} \frac{1}{n} \log p(\theta | \theta_{\text{ft}}^{(i)}, I)$. Using a Laplace approximation, the posterior for each $\theta_{\text{ft}}^{(i)}$ is approximated as a Gaussian centered at $\theta_{\text{ft}}^{(i)}$ with a precision matrix given by the Fisher Information Matrix $F_i$. By further approximating $F_i$ with its diagonal, Fisher Merging reduces to a weighted averaging of the finetuned parameters, resulting in the closed-form solution: $\theta_{\text{merged}} = \sum_{i=1}^{n} F_i \theta_{\text{ft}}^{(i)} / \sum_{i=1}^{n} F_i$.

**RegMean** [29] aims to align the activations of the merged model with those of the individual finetuned models. This is achieved by minimizing the Euclidean distance between activations in each linear layer produced by the merged model and the finetuned model. For the parameters in each linear $W^{(j)}$, Reg-Mean computes the merged result as $W_{\text{merged}}^{(j)} = \left( \sum_{i=1}^{n} X_i^{(j)\top} X_i^{(j)} \right)^{-1} \sum_{i=1}^{n} \left( X_i^{(j)\top} X_i^{(j)} W_i^{(j)} \right)$, where $X_i^{(j)}$ is the input features of the linear layers. The inner product matrix term is scaled by a hyperparmeter $\alpha$ to avoid degenerated solutions. For non-linear parameters such as those involved in attention computation, it applies simple averaging.

**TIES Merging** [75] introduces a three-step pipeline to resolve task interference during model merging: **i) Trim**: discard small-magnitude values in task vectors. **ii) Elect sign**: select the dominant sign for each parameter position, determined by whether the parameter has a higher total magnitude in the positive or negative direction. **iii) Merge**: combine model weights by retaining only parameters aligned with the elected sign. This process reduces destructive interference during merging.

**DARE** [83] performs random dropout on the task vectors based on the per-task binary mask $m_i$ drawn from the Bernouli distribution $m \sim \text{Bernouli}(p)$, where $p$ is a predefined dropout rate. To retain the original scale, the dropout task vectors are rescaled by $1/(1-p)$. Overall, the resulting sparse task vectors are $\tau_i = (1 - m_i) \odot \tau_i / (1-p)$, and then DARE proceeds with the task arithmetic procedure to combine task vectors.

**Consensus TA** [68] computes multi-task task vectors: $\tau_{\text{MTL}} = \theta_{\text{merged}} - \theta_{\text{pre}}$ with $\theta_{\text{merged}}$ obtained by task arithmetic. For each task $i$, it constructs a task-specific binary mask: $m_i = \mathbb{1}\{|\tau_i| \geq |\tau_{MTL} - \tau_i| \cdot \lambda_i\}$, where $\lambda_i$ is a tunable hyperparameter controlling the selectivity of task-specific information extraction. A final consensus mask selects parameters agreed upon by at least two tasks: $m_{\text{consensus}} = \mathbb{1}\{\sum_{i \in [n]} m_i \geq 2\}$. This mask is applied to the multi-task vector to filter out task-specific noise, producing a merged model that emphasizes parameter updates shared across multiple tasks.

**Localize-and-Stitch** [22] approaches sparsity differently by training binary masks that identify the most relevant parameters for each task. When there is no training data available, the algorithm has a dataless version which keeps the top-$k$ parameters in the task vectors with the largest magnitude. Only the localized regions of each model are stitched back onto the pretrained backbone. Unlike previous methods, it *does not* tune merging coefficients but simply averages selected regions with normalized coefficients.

While our benchmark encompasses a diverse array of model merging algorithms, it is not exhaustive. The field is rapidly evolving, with new methods continually emerging. We aim to expand our supported algorithms over time. Certain model merging approaches were not included in our benchmark due to compatibility issues or differing methodologies. For instance, AdaMerging [78] treats merging coefficients as trainable parameters, optimizing them via entropy minimization on unlabeled test data. While effective in vision tasks, its effectiveness is not tested on language models, as entropy minimization can lead to overconfident predictions and increased hallucinations. Similarly, Task Arithmetic in the Tangent Space [43] requires fine-tuning models within their tangent space, leveraging linear approximations to enhance weight disentanglement. This approach, though

theoretically sound, necessitates access to the fine-tuning process and may not be directly applicable in scenarios where only the final model checkpoints are available.

## B  Datasets Details

### B.1  Training data details

We present the training data statistics in Table 5.

Table 5: Datasets used for model training.

| Category | Dataset | Training method | # Data |
|---|---|---|---|
| Instruction-following | TULU-3 persona IF [32] | SFT | 29.9K |
| Mathematics | DART-Math [65] | SFT | 591K |
| | NuminaMath-TIR [34] | GRPO | 72.4K |
| Multilingual understanding | Aya [59] | SFT | 5.94K |
| Coding | Magicoder [69] | SFT | 110K |
| Safety | WildGuardMix [20] | SFT | 86.76K |
| | WildJailbreak [27] | SFT | 261.56K |

### B.2  Surrogate tasks for validation

Algorithms including Task Arithmetic, TIES Merging, DARE, Consensus TA and Dataless Localize-and-Stitch require additional validation data for hyperparameter tuning. However, in our evaluation suite, most tasks do not provide a dedicated validation split, as the available data is typically reserved entirely for evaluation. To address this, we select surrogate tasks that serve a similar purpose for each target category. Specifically, we use IFEval-like data [73] for instruction following validation, stem questions in MMLU [23] for math validation, CoNaLa [82] for coding validation, LAMBADA [46] for multilingual understanding validation, Wildjailbreak [27] for safety validation. These surrogate tasks provide practical alternatives for tuning hyperparameters while maintaining alignment with the goals of each specialized evaluation category.

### B.3  Licenses

NuminaMath-TIR, Aya and IFEval are under Apache 2.0 License. DART-Math, Magicoder, GSM8k, MATH and M_MMLU are under MIT license. XSTest, M_ARC and M_Hellaswag are under CC-BY-NC-4.0 License. TULU-3, WildGuardMix and WildJailbreak are under ODC-BY License. HumanEval+ and MBPP+ are under Apache License.

Llama-3.1 is under Llama 3.1 Community License Agreement and Llama-3.2 is under Llama 3.2 Community License Agreement. Gemma-2 is under Gemma Terms of Use.

## C  Implementation Details

### C.1  Training details

Table 6: Comparison of performance across tasks for different model sizes.

| Model Size | Instruction Following | Math | Multilingual | Coding | Safety |
|---|---|---|---|---|---|
| 2–3B models | 10 | 40 | 5 | 18 | 24 |
| 7–8B models | 36 | 154 | 17 | 66 | 82 |

For 2B and 3B models, we conduct experiments on NVIDIA RTX A6000 GPUs using a sequence length of 4096 and an initial learning rate of 1e-5. For 8B and 9B models, we use NVIDIA A100 GPUs with a sequence length of 3072 and an initial learning rate of 5e-5. Across all model sizes, we adopt the AdamW optimizer [39] with a cosine learning rate schedule, and set the global batch size to 128. For all tasks, we perform SFT for 2 epochs. We report the training cost in GPU hours for each task and model size in Table 6.

## C.2 Hyperparameter Tuning

Table 7: Hyperparameter tuning requirements for different algorithms.

| Algorithm | # Hyperparameter (combinations) | Hyperparameters |
|---|---|---|
| Model soup | 0 | - |
| Task arithmetic | 10 | scaling coef $\lambda \in \{0.1, 0.2, \cdots, 1\}$ |
| Fisher merging | 0 | - |
| RegMean | 5 | reduction $\alpha \in \{0.1, 0.3, 0.5, 0.7, 0.9\}$ |
| TIES | 30 | sparsity $s \in \{0.1, 0.2, 0.3\}$, scaling coef $\lambda \in \{0.1, 0.2, \cdots, 1\}$ |
| DARE | 30 | sparsity $s \in \{0.1, 0.2, 0.3\}$, scaling coef $\lambda \in \{0.1, 0.2, \cdots, 1\}$ |
| Consensus TA | 35 | sparsity $s \in \{0.2, 0.3, \cdots, 0.6\}$, scaling coef $\lambda \in \{0.1, 0.2, \cdots, 1\}$ |
| Dataless Localize-and-Stitch | 5 | sparsity $s \in \{0.1, 0.2, \cdots, 0.5\}$ |
| Localize-and-Stitch | 0 | - |

Merging algorithms have different requirements of hyperparameter tuning. We detail them in Table 7 following the practice specified in the original papers. For **RegMean**, although the original paper does not explicitly require hyperparameter tuning, we find that the selection of the scaling factor for the non-diagonal items in the inner product matrices dramatically influences the performance, and using the suggested $\alpha = 0.9$ often results in poor performance. Thus, we treat it as a hyperparameter and perform validation. For **Consensus TA**, each of the 5 tasks require tuning of the sparsity parameters, and subsequently, it requires tuning the scaling coefficients, totaling 35 runs. For **Dataless Localize-and-Stitch**, the original paper suggests a sparsity of $5\% \sim 10\%$. However, at larger model scales, we find that effective localization requires activating more than 10% of the parameters. This may be explained by the observation that larger models tend to distribute knowledge more broadly across their parameters [2, 7], making task-specific information less concentrated. Consistent with this observation, Poppi et al. [49] report that identifying safety-relevant regions in multilingual LLMs requires localizing up to 20% of the parameters. These findings suggest that the sparsity requirements for effective merging may scale with model size. Thus, we treat sparsity as a hyperparameter and search from 10% to 50%.

# D Discussions and Limitations

Here we discuss about the scope of this project and limitations of existing model merging literature.

Firstly, our evaluation is limited to merging models finetuned from the same initialization. Merging models from different base models is beyond the scope of our work, as it introduces fundamentally different challenges from the model merging setting we study. In our definition, model merging refers to the technique that uses arithmetic operations in the model parameter space to combine the strengths of multiple models. This formulation is widely adopted in the literature, and is highly valuable in practical scenarios where training data is inaccessible and multiple teams finetune the same model in parallel. In contrast, merging across model families requires tackling architectural and tokenization mismatches, where parameter-level arithmetic is not directly applicable. Prior works have explored combining knowledge from heterogeneous models, such as model routing that dynamically selects among models at inference time [77], or model ensembling that aggregates outputs [81]. While important in their own right, these directions constitute separate problem formulations that are not directly comparable to our work. While extending merging to heterogeneous base models is a promising direction, we believe the within-family merging problem remains a rich and impactful domain with immediate utility.

Secondly, we focus on merging dense models, without considering Mixture-of-Experts (MoE) models. Merging MoE models introduces challenges fundamentally different from dense model merging due to their sparse activation pattern. A key issue is expert index mismatch: different MoE models may assign distinct meanings to the same expert index, and merging without alignment disrupts the routing semantics. As a result, merging of experts or router weights can misroute inputs to inappropriate experts, leading to degraded performance. Thus, dense-model merging techniques are not readily applicable to MoEs. Instead, existing approaches construct new MoE architectures by combining dense experts and routing among them [35, 60], which lies beyond our definition of model merging.

Thirdly, our analysis and insights are mainly drawn from empirical observations, and it remains unclear whether the arguments can be tested theoretically. Theoretical analysis of model merging

is particularly challenging due to the scale and nonlinearity of modern transformer models, as well as the reliance on task-specific hyperparameter tuning. As a result, most progress in this area has been empirical, including our MergeBench, which is designed to systematically evaluate merging methods at scale. That said, we note that recent theoretical work has begun to analyze core components of model merging. For example, Li et al. [33] provides provable generalization guarantees for task vectors, showing that both low-rank approximation and magnitude-based pruning preserve performance, and that carefully chosen merging coefficients lead to strong generalization. These results support our empirical findings on the effectiveness of sparsification and coefficient tuning. only with the few attempts from linear mode connectivity [15], tangent task spaces [43] and task relationships [33, 85].

# E   Full Numeric Results

**Overall performance.** We report the full numeric results of the multi-task performance of each merging algorithm in Table 8 and generalization performance in Figure 3. As shown in Table 8, merging Gemma models often yields stronger results than merging Llama models of similar size. While the precise cause remains unclear due to limited transparency into pretraining procedures, this suggests that certain model architectures or training pipelines may be inherently more merging friendly. This is an interesting direction for further investigation.

Table 8: Average normalized multi-task performance on five categories for all models. The columns are sorted by the overall performance.

|  | Fisher Merging | DARE | Model soup | TIES | RegMean | Task arithmetic | Consensus TA | Dataless L&S | L&S |
|---|---|---|---|---|---|---|---|---|---|
| Gemma-2-2b | 68.4 | 66.1 | 66.4 | 61.7 | 76.8 | 70.3 | 74.7 | 76.1 | 76.8 |
| Gemma-2-2b-it | 88.8 | 84.3 | 89.9 | 87.0 | 91.6 | 89.9 | 90.5 | 90.6 | 90.4 |
| Llama-3.2-3B | 61.8 | 69.4 | 57.0 | 63.0 | 80.7 | 72.9 | 75.5 | 68.3 | 74.6 |
| Llama-3.2-3B-Instruct | 92.1 | 95.9 | 94.4 | 98.6 | 97.6 | 93.9 | 91.6 | 95.9 | 96.1 |
| Gemma-2-9b | 89.4 | 89.6 | 89.4 | 92.1 | 89.3 | 91.1 | 87.2 | 91.2 | 92.7 |
| Gemma-2-9b-it | 98.1 | 91.8 | 98.3 | 101.1 | 88.2 | 104.6 | 106.5 | 105.0 | 100.1 |
| Llama-3.1-8B | 80.3 | 78.7 | 81.1 | 81.4 | 80.6 | 84.8 | 83.5 | 90.6 | 91.7 |
| Llama-3.1-8B-Instruct | 85.4 | 88.8 | 88.6 | 93.7 | 90.4 | 90.0 | 91.1 | 95.2 | 95.1 |
| Overall | 83.0 | 83.1 | 83.1 | 84.8 | 86.9 | 87.2 | 87.6 | 89.1 | 89.7 |

Table 9: Average normalized generalization performance. The columns are sorted by the overall performance.

|  | Fisher Merging | DARE | RegMean | L&S | Task arithmetic | Dataless L&S | Consensus TA | Model soup | TIES |
|---|---|---|---|---|---|---|---|---|---|
| Gemma-2-2b | 99.3 | 95.5 | 99.8 | 96.3 | 91.0 | 96.3 | 95.0 | 99.3 | 99.5 |
| Gemma-2-2b-it | 101.2 | 101.5 | 102.7 | 100.3 | 101.6 | 100.3 | 102.7 | 102.5 | 102.6 |
| Llama-3.2-3B | 97.5 | 94.3 | 97.8 | 92.3 | 88.4 | 91.3 | 84.9 | 97.3 | 97.7 |
| Llama-3.2-3B-Instruct | 92.3 | 89.6 | 92.8 | 93.7 | 89.3 | 93.7 | 92.1 | 93.3 | 93.2 |
| Gemma-2-9b | 59.7 | 74.9 | 73.2 | 73.3 | 75.2 | 75.5 | 78.9 | 79.5 | 81.9 |
| Gemma-2-9b-it | 70.3 | 83.9 | 86.9 | 87.4 | 100.0 | 93.6 | 101.3 | 93.1 | 93.6 |
| Llama-3.1-8B | 77.0 | 73.4 | 77.3 | 74.9 | 68.5 | 74.0 | 69.2 | 77.4 | 80.8 |
| Llama-3.1-8B-Instruct | 87.9 | 83.9 | 71.9 | 84.2 | 91.9 | 93.8 | 94.4 | 85.6 | 83.9 |
| Overall | 85.7 | 87.1 | 87.8 | 87.8 | 88.3 | 89.8 | 89.8 | 91.0 | 91.7 |

**Detailed per-task performance.** To facilitate reproducibility of our evaluation results, we further report detailed per-task performance for all 8 models we test.

Table 10: Gemma-2-2B per-task and average results. Values are percentages.

| Task | Model soup | Task arithmetic | Fisher Merging | RegMean | TIES | DARE | Consensus TA | Dataless L&S | L&S |
|---|---|---|---|---|---|---|---|---|---|
| Instruction following | 19.6 | 29.4 | 22.6 | 24.6 | 19.8 | 26.3 | **26.3** | 17.0 | 23.1 |
| Math | 25.2 | 28.2 | 30.3 | 27.9 | 26.3 | 26.9 | 27.6 | **37.9** | 37.1 |
| Multilingual | 47.9 | 47.9 | 41.1 | 48.2 | 48.2 | **48.3** | **48.3** | 47.5 | 47.4 |
| Coding | 30.3 | **35.2** | 28.0 | 31.4 | 30.4 | 33.3 | 33.3 | 33.5 | 33.1 |
| Safety | 52.4 | 45.1 | 58.8 | 56.8 | 38.4 | 39.8 | 61.9 | **65.0** | 61.9 |
| Avg. Acc | 35.1 | 37.2 | 36.2 | **40.6** | 32.6 | 34.9 | 39.5 | 40.2 | **40.6** |
| Avg. Norm | 66.4 | 70.3 | 68.4 | **76.8** | 61.7 | 66.1 | 74.7 | 76.1 | **76.8** |

Table 11: Gemma-2-2B-IT per-task and average results. Values are percentages.

| Task | Model soup | Task arithmetic | Fisher Merging | RegMean | TIES | DARE | Consensus TA | Dataless L&S | L&S |
|---|---|---|---|---|---|---|---|---|---|
| Instruction following | 51.9 | 51.9 | 53.0 | 54.2 | 49.2 | 46.4 | **54.9** | 48.8 | 49.2 |
| Math | 38.7 | 38.7 | 39.7 | 40.5 | 38.5 | 36.7 | 38.4 | **47.9** | 47.9 |
| Multilingual | 49.2 | 49.2 | 48.7 | 48.7 | **49.3** | 49.0 | 49.2 | 48.8 | 48.7 |
| Coding | 40.2 | 40.2 | 40.1 | 39.3 | 39.6 | 38.3 | 39.7 | **40.2** | 38.2 |
| Safety | 81.3 | 81.3 | 76.8 | **83.6** | 76.3 | 74.8 | 81.0 | 79.8 | 79.0 |
| Avg. Acc | 52.3 | 52.3 | 51.7 | 53.3 | 50.6 | 49.0 | 52.7 | **52.7** | 52.6 |
| Avg. Norm | 89.9 | 89.9 | 88.8 | **91.6** | 87.0 | 84.3 | 90.5 | 90.6 | 90.4 |

Table 12: Llama-3.2-3B per-task and average results. Values are percentages.

| Task | Model soup | Task arithmetic | Fisher Merging | RegMean | TIES | DARE | Consensus TA | Dataless L&S | L&S |
|---|---|---|---|---|---|---|---|---|---|
| Instruction following | 7.2 | 25.3 | 12.0 | 14.2 | 9.6 | 18.7 | **30.5** | 10.4 | 23.7 |
| Math | 16.2 | 27.7 | 26.6 | 33.8 | 26.6 | 27.1 | 27.6 | **41.1** | 38.9 |
| Multilingual | 46.8 | 47.0 | 47.6 | **48.4** | 47.6 | 47.5 | 46.9 | 46.7 | 47.0 |
| Coding | 37.0 | **41.1** | 37.2 | 39.3 | 37.6 | 40.2 | 40.7 | 40.0 | 40.7 |
| Safety | 39.2 | 46.1 | 35.4 | **71.7** | 40.4 | 44.9 | 48.2 | 37.3 | 41.3 |
| Avg. Acc | 29.3 | 37.5 | 31.8 | **41.5** | 32.3 | 35.7 | 38.8 | 35.1 | 38.3 |
| Avg. Norm | 57.0 | 72.9 | 61.8 | **80.7** | 63.0 | 69.4 | 75.5 | 68.3 | 74.6 |

Table 13: Llama-3.2-3B-Instruct per-task and average results. Values are percentages.

| Task | Model soup | Task arithmetic | Fisher Merging | RegMean | TIES | DARE | Consensus TA | Dataless L&S | L&S |
|---|---|---|---|---|---|---|---|---|---|
| Instruction following | 56.0 | **59.7** | 53.6 | 56.9 | 56.6 | 48.6 | 58.8 | 55.8 | 57.2 |
| Math | 53.9 | 55.1 | 49.8 | **61.0** | 56.7 | 56.7 | 51.1 | 59.8 | 59.9 |
| Multilingual | 45.0 | 45.2 | 44.7 | **48.4** | 44.6 | 43.9 | 45.1 | 44.1 | 44.2 |
| Coding | 52.4 | 49.8 | 51.2 | 51.8 | 52.5 | **53.2** | 48.0 | 51.8 | 52.8 |
| Safety | 84.6 | 80.6 | 85.3 | 87.5 | **94.5** | 94.0 | 80.0 | 85.1 | 83.1 |
| Avg. Acc | 58.4 | 58.1 | 56.9 | 60.3 | 60.9 | 59.3 | 56.6 | 59.3 | **59.5** |
| Avg. Norm | 94.4 | 93.9 | 92.1 | 97.6 | 98.6 | 95.9 | 91.6 | 95.9 | **96.1** |

Table 14: Gemma-2-9B per-task and average results. Values are percentages.

| Task | Model soup | Task arithmetic | Fisher Merging | RegMean | TIES | DARE | Consensus TA | Dataless L&S | L&S |
|---|---|---|---|---|---|---|---|---|---|
| Instruction following | 30.3 | 31.2 | 27.5 | 33.8 | 28.8 | 32.0 | 23.7 | 30.5 | **35.3** |
| Math | 60.3 | 64.5 | 48.2 | 59.9 | 65.3 | 62.8 | 59.0 | **67.2** | 66.5 |
| Multilingual | **60.0** | 57.1 | 58.8 | 55.2 | 59.5 | 56.5 | 59.5 | 56.5 | 55.1 |
| Coding | 51.5 | 50.8 | 56.6 | 39.2 | 52.3 | 48.8 | 51.4 | **56.0** | 53.3 |
| Safety | 70.6 | 74.4 | 81.7 | **84.4** | 75.3 | 73.2 | 72.6 | 68.1 | 72.6 |
| Avg. Acc | 54.6 | 55.6 | 54.5 | 54.5 | 56.2 | 54.7 | 53.2 | 55.7 | **56.6** |
| Avg. Norm | 89.4 | 91.1 | 89.3 | 89.3 | 92.1 | 89.5 | 87.2 | 91.2 | **92.6** |

Table 15: Gemma-2-9B-IT per-task and average results. Values are percentages.

| Task | Model soup | Task arithmetic | Fisher Merging | RegMean | TIES | DARE | Consensus TA | Dataless L&S | L&S |
|---|---|---|---|---|---|---|---|---|---|
| Instruction following | 50.5 | 59.3 | 54.2 | 47.5 | 52.9 | 44.2 | **62.5** | 60.6 | 57.3 |
| Math | 64.4 | 64.3 | 55.5 | 52.4 | 66.3 | 62.5 | 63.8 | **70.4** | 67.7 |
| Multilingual | 60.9 | 63.0 | 58.6 | 49.5 | 60.6 | 57.6 | 63.3 | **63.3** | 57.7 |
| Coding | 58.5 | 59.8 | 58.9 | 48.8 | 59.5 | 55.7 | **60.5** | 59.4 | 55.6 |
| Safety | 68.2 | 75.3 | 74.7 | 73.1 | 71.6 | 62.4 | 77.3 | **79.0** | 69.7 |
| Avg. Acc | 60.5 | 64.4 | 60.4 | 54.2 | 62.2 | 56.5 | **65.5** | 64.6 | 61.6 |
| Avg. Norm | 98.3 | 104.6 | 98.1 | 88.2 | 101.0 | 91.8 | **106.5** | 104.9 | 100.1 |

Table 16: Llama-3.1-8B per-task and average results. Values are percentages.

| Task | Model soup | Task arithmetic | Fisher Merging | RegMean | TIES | DARE | Consensus TA | Dataless L&S | L&S |
|---|---|---|---|---|---|---|---|---|---|
| Instruction following | 8.3 | 31.2 | 5.2 | 10.9 | 12.2 | 13.1 | 26.3 | 18.1 | **37.3** |
| Math | 50.1 | 55.5 | 48.0 | 52.9 | 56.3 | 55.6 | 54.2 | **59.5** | 57.0 |
| Multilingual | 54.0 | 49.1 | 52.1 | 51.9 | **54.5** | 52.7 | 49.0 | 52.0 | 54.3 |
| Coding | 49.6 | 48.8 | 49.5 | 47.9 | 49.0 | 49.3 | 49.6 | **51.3** | 50.9 |
| Safety | 71.0 | 59.0 | 76.0 | 67.8 | 61.9 | 55.4 | 60.7 | **79.3** | 63.7 |
| Avg. Acc | 46.6 | 48.7 | 46.2 | 46.3 | 46.8 | 45.2 | 48.0 | 52.0 | **52.7** |
| Avg. Norm | 81.1 | 84.8 | 80.3 | 80.6 | 81.4 | 78.7 | 83.5 | 90.6 | **91.7** |

Table 17: Llama-3.1-8B-Instruct per-task and average results. Values are percentages.

| Task | Model soup | Task arithmetic | Fisher Merging | RegMean | TIES | DARE | Consensus TA | Dataless L&S | L&S |
|---|---|---|---|---|---|---|---|---|---|
| Instruction following | 37.5 | 47.0 | 31.8 | 46.8 | 43.4 | 39.8 | 48.4 | **55.4** | 44.4 |
| Math | 64.4 | 60.3 | 52.3 | 59.9 | 65.7 | 63.5 | 63.3 | **68.2** | 67.1 |
| Multilingual | 53.6 | **54.8** | 54.3 | 50.9 | 53.9 | 51.4 | 54.7 | 54.2 | 52.3 |
| Coding | 62.1 | 61.8 | **63.6** | 58.0 | 62.6 | 57.3 | 61.2 | 62.9 | 62.0 |
| Safety | 81.4 | 79.8 | 86.2 | 89.4 | 90.4 | 87.8 | 79.6 | 80.4 | **91.8** |
| Avg. Acc | 59.8 | 60.7 | 57.6 | 61.0 | 63.2 | 59.9 | 61.4 | **64.2** | 63.5 |
| Avg. Norm | 88.6 | 90.0 | 85.4 | 90.4 | 93.7 | 88.8 | 91.1 | **95.2** | 94.1 |

