# OpenReview forum: "MergeBench: A Benchmark for Merging Domain-Specialized LLMs"
_NeurIPS.cc/2025/Datasets_and_Benchmarks_Track — NeurIPS 2025 Datasets and Benchmarks Track poster_

### Official Review · Reviewer_buXu · 2025-07-01

**Rating:** 4
**Confidence:** 5

**Summary:**

This paper introduces MergeBench, a benchmark designed to evaluate the performance differences of various model merging techniques for large language models. To this end, fine-tuned versions of different models are used to construct the benchmark across a variety of tasks. This study aims to address the limitations of existing model merging evaluation methods in terms of model scale and task diversity.

**Additional Feedback:**

Please refer to Weaknesses 1–4 and the question below.

1. Have the authors considered constructing benchmarks for vision models, multimodal models, etc.? And do you believe that these models would exhibit different characteristics compared to LLMs?

If you can address these issues, I will consider raising the score.

**Dataset Code Accessibility:**

Yes

**Dataset Code Comments:**

The paper provides a GitHub repository that includes the data and code.

**Ethical Comments:**

The paper uses only open-source datasets, and all base models are also open-source. The tasks evaluated are commonly used in the industry, so there are no ethical concerns involved.

**Ethical Considerations:**

No, there are no or only very minor ethics concerns

**Final Justification:**

The author's response has addressed most of my concerns. This paper presents a benchmark specifically for LLM merging, and in fact, my recommendation regarding its acceptance is entirely neutral. That said, I believe it would become a strong paper if the authors include more models and tasks.

**Limitations Weaknesses:**

1. The current selection of models remains relatively limited, including only four models from the Llama and Gemma families. To better reflect the diversity of the benchmark, it would be beneficial to include more model families such as Qwen, as well as larger-scale models.

2. The current analysis mainly focuses on comparisons between different model merging methods and between base and instruction-tuned models, which makes the overall analysis somewhat superficial. More detailed insights could be added. For example, why do different merging methods show smaller performance differences when applied to instruction-tuned models compared to base models? Why does the performance gain after merging instruction-tuned models appear relatively small compared to the original models? Additionally, it would be valuable to explore whether models from different families exhibit distinct behaviors during the merging process.

3. During the model merging process, hyperparameters can significantly influence the final results. How does the paper ensure that the selected hyperparameters are close to optimal? The proposed solution in the paper is to use validation tasks. However, can this approach guarantee that the selected hyperparameters will perform well during testing? Furthermore, can the paper provide users with practical guidance or references for choosing effective hyperparameters?

4. While the paper presents averaged performance across different tasks, including a more detailed per-task analysis would make the evaluation more compelling. It would also be informative to investigate whether different tasks exhibit varying preferences for specific model merging techniques.

**Strengths Contributions:**

1. MergeBench addresses the limitations of previous model merging evaluations in terms of model scale and task diversity, making it a more relevant and robust benchmark for assessing current LLM merging techniques.

2. The benchmark suite covers five key task domains including instruction following, mathematics, multilingual understanding, coding, and safety, and employs state-of-the-art open-source LLMs, including the Llama and Gemma families, with parameter scales ranging from 2B to 9B. This broad coverage provides a more thorough evaluation of model merging techniques.

3. Some analysis has been conducted on the results of model merging. For example, fine-tuning based on the base model leads to greater improvements compared to the instruction-tuned model. Additionally, various existing model merging methods are compared within the same benchmark, offering valuable performance references for future researchers.

---

> ### Author Rebuttal · Authors · 2025-07-31
>
> **[W1- More model families]** To broaden the coverage of MergeBench, we additionally include experiments on Qwen3-4B. Due to the limited time in the rebuttal phase, we evaluate a single base model, but we plan to continually expand model coverage in future updates.
>
> Below we present the normalized per-task performance of various merging algorithms on Qwen3-4B. Consistent with our findings on other instruction-tuned models (line 185), performance differences across methods are smaller, with most approaches achieving over 90% normalized performance. Both variants of Localize-and-Stitch remain the strongest performers, especially on tasks like mathematics.
>
> |                       | Model soup | Task arithmetic | RegMean | TIES   | DARE       | Consensus TA | Dataless Localize-and-Stitch | Localize-and-Stitch |
> |-----------------------|------------|-----------------|---------|--------|------------|--------------|------------------------------|---------------------|
> | Instruction following |     0.7236 |          0.7298 |  0.6164 | 0.6708 |     0.7019 |       0.7236 |                   **0.8727** |              0.7391 |
> | Math                  |     0.9780 |          0.9930 |  0.8843 | 0.9578 |     0.9468 |       0.9763 |                   **0.9974** |              0.9922 |
> | Multilingual          |     0.9239 |          0.9397 |  0.8485 | 0.9102 |     0.9329 |       0.9242 |                   **0.9743** |              0.9271 |
> | Coding                |     0.9724 |          0.9926 |  0.8913 | 0.9443 | **0.9975** |       0.9757 |                       0.9924 |              0.9953 |
> | Safety                |     0.9790 |          0.9809 |  0.9149 | 0.9833 |     0.9857 |       0.9819 |                       1.0010 |          **1.0102** |
> | Avg. Norm             |     0.9154 |          0.9272 |  0.8311 | 0.8933 |     0.9130 |       0.9163 |                   **0.9676** |              0.9328 |
>
> ---
>
> **[W2 - Detailed insights]**
> - **Instruction-tuned (IT) v.s. Pretrained models:** Firstly, from a model training perspective, the IT models go through further training compared to the pretrained ones, which enhances their ability to follow instructions and produce formatted outputs aligned with user expectations. This inherently leaves less room for improvement through further post-training on certain tasks. For example, we find that further finetuning on instruction-following datasets often yields limited gains on IT models. Secondly, from a model merging perspective, prior works [1,2] suggest that large-scale instruction tuning helps disentangle model weights, which facilitates merging. This likely explains the smaller performance gap between merging methods on IT models.
> - **Different model families:** The differences between different model families largely rely on the quality of base models. As shown in Table 7 of Appendix D, merging Gemma models often yields stronger results than merging Llama models of similar size. While the precise cause remains unclear due to limited transparency into pretraining procedures, this suggests that certain model architectures or training pipelines may be inherently more merging friendly. This is an interesting direction for further investigation, and we will expand on these insights in the final manuscript.
>
> [1] What Matters for Model Merging at Scale? Yadav et al, 2024.
>
> [2] Task Arithmetic in the Tangent Space: Improved Editing of Pre-Trained Models. Ortiz-Jimenez et al, 2023.
>
> ---
>
> **[W3- Hyperparameter]**
> - **Validation-based tuning:** As detailed in Appendix C.2, we closely follow the hyperparameter tuning strategies recommended by the original papers for each method, and **all of which rely on validation performance** [1,2,3,4]. However, this strategy faces a key challenge in LLM tuning as most LLM benchmarks are used in their entirety for evaluation and lack predefined validation splits. To address this, in Appendix B.2, we further detail our careful selection of surrogate validation datasets that exhibit strong correlation with test set performance. This ensures effective tuning without test data leakage.
> - **Practical guidance:** In Appendix C.2, we provide practical guidance on hyperparameter ranges. For instance, for Localize-and-Stitch, we observe that larger models require broader localized regions than originally suggested. For Regmean, we identify the importance of tuning the reduction parameter, which is a new discovery complementary to the original paper’s suggestion. However, for other hyperparameters like scaling factors, we find no universal simplification.  Thus, we mentioned on line 289 that hyperparameter tuning cost remains a key challenge in this field that requires significant effort for future improvement.
>
> [1] Editing Models with Task Arithmetic. Ilharco et al, 2023.
>
> [2] TIES-MERGING: Resolving Interference When Merging Models. Yadav et al, 2023.
>
> [3] Language Models are Super Mario: Absorbing Abilities from Homologous Models as a Free Lunch. Yu et al, 2024.
>
> [4] Localizing Task Information for Improved Model Merging and Compression. Wang et al, 2024.
>
> ---
>
> **[W4 - Per-task performance]** We have included the full per-task results for the Gemma-2-2B model below. Since tasks vary significantly in difficulty, we report normalized performance, which reflects the percentage of performance recovered relative to the individually fine-tuned models.
>
> We observe that both variants of Localize-and-Stitch consistently achieve the best performance on the math task across models. This could be an indication that mathematical skills may exhibit localized patterns particularly well-captured by these methods. Beyond this, we do not find consistent trends linking specific merging methods to particular task domains. We will include full per-task performance for all models in the final manuscript.
>
> |                       | Model soup | Task arithmetic | Fisher Merging | RegMean | TIES   | DARE       | Consensus TA | Dataless Localize-and-Stitch | Localize-and-Stitch |
> |-----------------------|------------|-----------------|----------------|---------|--------|------------|--------------|------------------------------|---------------------|
> | Instruction following |     0.4796 |      **0.7195** |         0.5520 |  0.6017 | 0.4842 |     0.6426 |       0.6425 |                       0.4163 |              0.5660 |
> | Math                  |     0.4913 |          0.5497 |         0.5910 |  0.5461 | 0.5131 |     0.5251 |       0.5392 |                   **0.7405** |              0.7246 |
> | Multilingual          |     0.9422 |          0.9406 |         0.7945 |  0.9422 | 0.9477 | **0.9499** |       0.9490 |                       0.9337 |              0.9302 |
> | Coding                |     0.8433 |      **0.9816** |         0.7795 |  0.8758 | 0.8462 |     0.9287 |       0.9270 |                       0.9335 |              0.9278 |
> | Safety                |     0.5812 |          0.5001 |         0.6515 |  0.6294 | 0.4254 |     0.4409 |       0.6866 |                   **0.7207** |              0.6863 |
>
> ---
>
> **[Q1 - Multimodal benchmarks]** There has been prior work on model merging benchmarks for vision models, primarily in the context of image classification with CLIP backbones [1,2]. As multimodal models become more capable, we agree that extending MergeBench to vision-language models is a promising direction, potentially introducing new challenges. Nevertheless, we expect their merging dynamics to remain largely similar to those of LLMs. This is because the vision encoder typically constitutes a small portion of the model—for example, Qwen-2-VL's vision tower is only 675M across all model sizes (1.5B–72B). Moreover, many vision-language training protocols [3,4] freeze the vision encoder and finetune only the language components, which dominate the model’s behavior. Thus, we anticipate that insights from LLM merging will largely carry over to VLMs.
>
> [1] Fusionbench: A comprehensive benchmark of deep model fusion. Tang et al, 2024.
>
> [2] Realistic evaluation of model merging for compositional generalization. Tam et al, 2024.
>
> [3] Qwen2-VL: Enhancing Vision-Language Model’s Perception of the World at Any Resolution. Qwen Team, 2024.
>
> [4] AGUVIS: Unified Pure Vision Agents for Autonomous GUI Interaction. Xu et al, 2025.

---

> > ### Comment · Reviewer_buXu · 2025-08-05
> >
> > Thank you to the author for responding to my question. This has addressed most of my concerns. I still have one question: In the rebuttal, what training data and training methodology were used for the merging of qwen3-4b? In addition, could you provide the performance of qwen3-4b on these tasks without training and merging?

---

> > ### Author Response · Authors · 2025-08-05
> >
> > Thank you for your follow-up. We are glad that our previous response addressed most of your concerns. Regarding your remaining question:
> >
> > **[Training for Qwen]** For Qwen3-4B, we used the same training datasets and methodology as in our Llama and Gemma experiments. The datasets are listed in Table 2, and the training procedure is described in Section 3.3 of the main paper.
> >
> > **[Base Qwen performance]** Below, we report the performance of Qwen3-4B before and after SFT. The SFT column reflects the performance of models individually finetuned on the corresponding task’s training data. The values in parentheses indicate the normalized performance (i.e., base model performance divided by its SFT counterpart), which helps account for varying task difficulty. As shown, SFT consistently improves performance across all tasks, with instruction following showing the largest relative gain and multilingual the smallest. In addition, the base model's performance falls short of merged models in general. We will include these results in the final version of the manuscript.
> >
> >
> > |                       | Qwen3-4B perf. (normalized)        | SFT perf.    |
> > |-----------------------|-----------------|--------|
> > | Instruction following | 0.3956 (0.6646) | 0.5952 |
> > | Math                  | 0.6298 (0.8972) | 0.7020 |
> > | Multilingual          | 0.5284 (0.9263) | 0.5704 |
> > | Coding                | 0.5552 (0.8775) | 0.6327 |
> > | Safety                | 0.6726 (0.8029) | 0.8377 |
> > | Avg                   | 0.5563 (0.8333) | 0.6676 |

---

> > > ### Author Response · Authors · 2025-08-08
> > >
> > > Dear reviewer buXu,
> > >
> > > We want to sincerely thank you for the valuable insights and great questions. We work deligently during the rebuttal period to provide additional results on the Qwen models and report the detailed per-task performance. As the author-reviewer discussion period ends **tomorrow**, we are more than willing to follow up if you have any remaining concerns. If all the questions are resolved, we would greatly appreciate it if you could acknowledge our response.
> > >
> > > Thank you so much.
> > >
> > > Authors

---

> > > > ### Comment · Reviewer_buXu · 2025-08-08
> > > >
> > > > Thank you for the author's response. It has resolved most of my concerns. Therefore, I will raise my score to 4.

---

### Official Review · Reviewer_SK6G · 2025-07-01

**Rating:** 5
**Confidence:** 4

**Summary:**

The paper introduces MergeBench, the first large-scale, open benchmark designed to evaluate model-merging techniques for domain-specialised LLMs. Starting from eight recent open-source base models (Llama-3 and Gemma families, 2 B – 9 B)
, the authors finetune each on five disjoint skills—instruction following, mathematics, multilingual understanding, coding and safety
. They then apply eight representative merging algorithms covering coefficient-tuning and sparsification families.

**Dataset Code Accessibility:**

Yes

**Dataset Code Comments:**

There is a public code link to access. The author also provided rich experimental details in the appendix.

**Ethical Considerations:**

No, there are no or only very minor ethics concerns

**Final Justification:**

The authors have addressed my initial concerns. This work provides valuable empirical insights into LLM merging. I tend to accept it.

**Limitations Weaknesses:**

[1] Only Llama/Gemma backbones are considered; results may not transfer to Mistral, Qwen, or proprietary architectures.\
[2] All experiments merge models fine-tuned from one common initialization; how to merge models that originate from different base architectures is still unsolved.\
[3] All takeaways stem from empirical results, with no formal proofs or analytic guarantees provided to underpin the claims.

**Strengths Contributions:**

[1] Comprehensive, domain-diverse benchmark. Five task families chosen for real-world relevance and merge compatibility.\
[2] Systematic comparison of 8 algorithms spanning coefficient tuning and sparsification, with unified training/validation protocol.\
[3] Actionable empirical insights. Finds merging more effective on stronger or instruction-tuned bases and that sparsity/coefficient tuning curbs forgetting.\
[4] Table 4 clearly situates MergeBench relative to FusionBench, Model-GLUE, etc., highlighting unique coverage and openness.\
[5] The author has provided rich experimental details in the appendix, which is convenient for colleagues to use and reproduce.

---

> ### Author Rebuttal · Authors · 2025-07-31
>
> **[W1- More model families]** To broaden the coverage of MergeBench, we additionally include experiments on Qwen3-4B. Due to the limited time in the rebuttal phase, we evaluate a single base model, but we plan to continually expand model coverage in future updates.
>
> Below we present the normalized per-task performance of various merging algorithms on Qwen3-4B. Consistent with our findings on other instruction-tuned models (line 185), performance differences across methods are smaller, with most approaches achieving over 90% normalized performance. Both variants of Localize-and-Stitch remain the strongest performers, especially on tasks like mathematics.
>
> |                       | Model soup | Task arithmetic | RegMean | TIES   | DARE       | Consensus TA | Dataless Localize-and-Stitch | Localize-and-Stitch |
> |-----------------------|------------|-----------------|---------|--------|------------|--------------|------------------------------|---------------------|
> | Instruction following |     0.7236 |          0.7298 |  0.6164 | 0.6708 |     0.7019 |       0.7236 |                   **0.8727** |              0.7391 |
> | Math                  |     0.9780 |          0.9930 |  0.8843 | 0.9578 |     0.9468 |       0.9763 |                   **0.9974** |              0.9922 |
> | Multilingual          |     0.9239 |          0.9397 |  0.8485 | 0.9102 |     0.9329 |       0.9242 |                   **0.9743** |              0.9271 |
> | Coding                |     0.9724 |          0.9926 |  0.8913 | 0.9443 | **0.9975** |       0.9757 |                       0.9924 |              0.9953 |
> | Safety                |     0.9790 |          0.9809 |  0.9149 | 0.9833 |     0.9857 |       0.9819 |                       1.0010 |          **1.0102** |
> | Avg. Norm             |     0.9154 |          0.9272 |  0.8311 | 0.8933 |     0.9130 |       0.9163 |                   **0.9676** |              0.9328 |
>
>
> ---
>
> **[W2 - Merging different base models]** Merging models from different base models is beyond the scope of our work, as it introduces fundamentally different challenges from the model merging setting we study. In our definition, model merging refers to the technique that uses arithmetic operations in the model parameter space to combine the strengths of multiple models. This formulation is widely adopted in the literature, and is highly valuable in practical scenarios where training data is inaccessible and multiple teams finetune the same model in parallel.
>
> In contrast, merging across model families requires tackling architectural and tokenization mismatches, where parameter-level arithmetic is not directly applicable. Prior works have explored combining knowledge from heterogeneous models, such as model routing that dynamically selects among models at inference time [1], or model ensembling that aggregates outputs [2]. While important in their own right, these directions constitute separate problem formulations that are not directly comparable to our work. While extending merging to heterogeneous base models is a promising direction, we believe the within-family merging problem remains a rich and impactful domain with immediate utility.
>
> [1] A Survey on Model MoErging: Recycling and Routing Among Specialized Experts for Collaborative Learning. Yadav et al, 2024.
>
> [2] Determine-Then-Ensemble: Necessity of Top-k Union for Large Language Model Ensembling. Yao et al, 2024.
>
>
>
> ---
>
> **[W3 - Theoretical insights]** Theoretical analysis of model merging is particularly challenging due to the scale and nonlinearity of modern transformer models, as well as the reliance on task-specific hyperparameter tuning. As a result, most progress in this area has been empirical, including our MergeBench, which is designed to systematically evaluate merging methods at scale. That said, we note that recent theoretical work has begun to analyze core components of model merging. For example, [1] provides provable generalization guarantees for task vectors, showing that both low-rank approximation and magnitude-based pruning preserve performance, and that carefully chosen merging coefficients lead to strong generalization. These results support our empirical findings on the effectiveness of sparsification and coefficient tuning. We will incorporate this connection in our final manuscript.
>
> [1] When is Task Vector Provably Effective for Model Editing? A Generalization Analysis of Nonlinear Transformers. Li et al, 2025.

---

### Official Review · Reviewer_9W6o · 2025-07-01

**Rating:** 4
**Confidence:** 2

**Summary:**

This paper introduces MergeBench, a comprehensive benchmark designed to evaluate model merging techniques for domain-specialized large language models (LLMs). The authors construct a suite of 40 specialized models based on open-source LLaMA and Gemma models ranging from 2B to 9B parameters, covering five diverse domains: instruction following, mathematics, multilingual understanding, coding, and safety. They evaluate eight representative merging methods along three dimensions: multi-task performance, forgetting (retention of pretrained knowledge), and runtime efficiency. The benchmark reveals that model merging performs better with larger and instruction-tuned base models, and that coefficient tuning and sparsification help mitigate forgetting. MergeBench is open-sourced, providing a foundation for future research.

**Dataset Code Accessibility:**

Yes

**Ethical Considerations:**

No, there are no or only very minor ethics concerns

**Final Justification:**

Thank the authors for the response. This is an interesting topic and I will maintain my recommendation for acceptance.

**Limitations Weaknesses:**

- Lines 129–131 mention the construction of specialized and instruction-tuned models, but the paper does not report the total training cost or compute budget required for building them.
- The claim that smaller models suffer more from task interference due to limited capacity (“This performance gap suggests…” in line 190) is plausible but not rigorously supported. It is unclear whether this is based on empirical measurements or theoretical justification, and the paper does not provide direct evidence for this assertion.
- In figure 3, if it is continual finetuning, what is the accuracy approximately? Because in fact, model merging also has a certain degree of accuracy reduction.
- The benchmark is limited to dense transformer-based models from the LLaMA and Gemma families. It does not evaluate the merging performance on newer architectures such as Mixture-of-Experts (MoE) models, which are becoming increasingly relevant in practice.

**Strengths Contributions:**

- The analysis in Sections 4.1 (Multi-Task Performance) and 4.2 (Retention of Base Model Knowledge) reveals interesting empirical findings not shown in prior work—for example, the improved merging performance on instruction-tuned models and the effectiveness of sparsity for mitigating forgetting.
- The authors built both domain-specialized models and their instruction-tuned versions across five tasks and multiple model sizes (LLaMA and Gemma, 2B–9B). This required significant engineering and computational effort.
- The discussion in Section 6 (“Mix data or merge models?”) raises thoughtful questions about the trade-offs between multi-task training and model merging, offering useful insights for future research.
- The benchmark is well-designed with careful control of model variants and task isolation, and includes both gradient-based and sparsification-based merging methods.

---

> ### Author Rebuttal · Authors · 2025-07-31
>
> **[W1 - Compute cost]** We conduct all training on NVIDIA RTX A6000 GPUs, and we report the training cost in GPU hours for each task and model size below. The numbers reflect the total compute required to train the specialized models used in our benchmark. We will include this information in the final manuscript.
> |             | Instruction following | Math | Multilingual | Coding | Safety |
> |:-----------:|:---------------------:|:----:|:------------:|:------:|:------:|
> | 2-3B models |           10          |  40  |       5      |   18   |   24   |
> | 7-8B models |           36          |  154 |      17      |   66   |   82   |
>
> ---
>
> **[W2 - Capacity limitation for small models]** From line 192-194, we further ground our observation on analogous insights from multi-task learning. In particular, [1] has theoretically proven that for over-parametrized multi-task models with sufficient width, the trade-off among tasks are minimal, and any convex combination of task objectives achieves optimal multi-task performance. In contrast, under-parameterized models face inherent trade-offs, making them more vulnerable to task interference. This theoretical result provides a rigorous basis for our empirical observation that smaller models exhibit more performance degradation under merging due to limited capacity. We will further clarify this connection in our final manuscript.
>
> [1] Revisiting Scalarization in Multi-Task Learning: A Theoretical Perspective. Hu et al, 2024.
>
> ---
>
> **[W3 - Continual finetuning]** We interpret the reviewer’s question as referring to the performance of sequentially finetuning on each task. We do not report this setting in the paper because its performance is highly sensitive to the order in which tasks are trained, and exhaustively evaluating all permutations is computationally infeasible.
>
> Nevertheless, prior work (e.g., Figure 9 in [1]) shows that continual finetuning suffers from catastrophic forgetting, where performance on earlier tasks degrades severely as the model adapts to later ones. As we note on line 212, model merging yields lower in-domain performance than multi-task training, and continual finetuning typically performs even worse due to this forgetting.
>
> In Figure 3, the x-axis reflects retention relative to individually finetuned models on the five categories of interest, and the y-axis reflects retention of pretrained knowledge relative to the base model. Based on prior findings, a continually finetuned model would likely fall in the lower-left region of the plot, indicating poor performance on both axes.
>
> [1] Localize-and-Stitch: Efficient Model Merging via Sparse Task Arithmetic. He et al, 2025.
>
> ---
>
> **[W4- MoE merging]**
> Merging MoE models introduces challenges fundamentally different from dense model merging due to their *sparse activation* pattern. A key issue is expert index mismatch: different MoE models may assign distinct meanings to the same expert index, and merging without alignment disrupts the routing semantics. As a result, merging of experts or router weights can misroute inputs to inappropriate experts, leading to degraded performance. Thus, dense-model merging techniques are not readily applicable to MoEs. Instead, existing approaches construct new MoE architectures by combining dense experts and routing among them [1,2], which lies beyond our definition of model merging.
>
> [1] Branch-Train-Merge: Embarrassingly Parallel Training of Expert Language Models. Li et al, 2022.
>
> [2] Branch-Train-MiX: Mixing Expert LLMs into a Mixture-of-Experts LLM. Sukhbaatar et al, 2024.

---

> > ### Author Response · Authors · 2025-08-08
> >
> > Dear reviewer 9W6o,
> >
> > We want to sincerely thank you for the valuable insights and great questions. We work deligently during the rebuttal period to provide explanation of the capacity limitation of small models, challenges of MoE merging and report compute cost. As the author-reviewer discussion period ends **tomorrow**, we are more than willing to follow up if you have any remaining concerns. If all the questions are resolved, we would greatly appreciate it if you could acknowledge our response.
> >
> > Thank you so much.
> >
> > Authors

---

> > ### Comment · Reviewer_9W6o · 2025-08-09
> >
> > Thank the authors for the response. This is an interesting topic and I will maintain my recommendation for acceptance.

---

### Official Review · Reviewer_dFEo · 2025-07-02

**Rating:** 5
**Confidence:** 3

**Summary:**

This paper introduces MergeBench, a comprehensive benchmark designed to rigorously evaluate model merging techniques for Large Language Models (LLMs). The authors argue that existing evaluations are limited by small model scales and narrow task diversity. To address this, MergeBench uses modern, open-source base models (Llama and Gemma families, up to 9B parameters) and creates specialized versions across five key domains: instruction following, mathematics, coding, multilingualism, and safety. The work standardizes the finetuning and evaluation pipeline to assess eight representative merging methods. Performance is measured across three dimensions: multi-task capability, knowledge retention (forgetting), and runtime efficiency. The key findings show that merging is more effective on stronger base models and that techniques like sparsification help mitigate forgetting, though challenges in computational cost and performance compared to multi-task training remain.

**Dataset Code Accessibility:**

Yes

**Ethical Considerations:**

No, there are no or only very minor ethics concerns

**Final Justification:**

I have no further concerns.

**Limitations Weaknesses:**

- The benchmark, while thorough, is confined to the "best-case" scenario of merging models that share an identical pretrained base. As acknowledged in the appendix, it does not address the more complex and often more practical challenge of merging models from different families or architectures (e.g., merging a Llama-based model with a Gemma-based one). This limits the generalizability of the findings and leaves a critical area of model merging unexplored, where architectural incompatibilities would likely surface entirely new challenges.

### Questions ###
- The term "MTL" in Figure 3's legend is not defined in the caption.
- Figure 5's caption claims a "favorable balance" for RegMean, but L&S appears strictly better.

**Strengths Contributions:**

- The primary strength of this work is the creation of a much-needed, comprehensive, and modern benchmark for model merging. The authors thoughtfully select diverse and relevant base models (Llama and Gemma) at a meaningful scale (up to 9B). The five task domains are well-chosen to represent distinct, practical capabilities beyond simple NLP tasks. By systematically creating their own specialized models from common bases, the authors ensure a controlled and fair comparison, a significant improvement over prior work that relied on smaller models or a less standardized collection of tasks.
- MergeBench provides a robust evaluation framework that goes beyond simple task performance. The inclusion of analyses on knowledge retention (forgetting) and runtime efficiency provides a holistic view of the practical trade-offs involved in different merging methods. The paper presents its findings clearly, with informative figures (e.g., Figure 3 on the generalization vs. in-domain trade-off) and a detailed breakdown of runtime costs (Figure 4). The "Practical guideline" in Section 4.3 is a particularly valuable contribution, offering clear, actionable advice for practitioners based on the empirical results.

---

> ### Author Rebuttal · Authors · 2025-07-31
>
> **[W1 - Merging different base model]** Merging models from different base models is beyond the scope of our work, as it introduces fundamentally different challenges from the model merging setting we study. In our definition, model merging refers to the technique that uses arithmetic operations in the model parameter space to combine the strengths of multiple models. This formulation is widely adopted in the literature, and is highly valuable in practical scenarios where training data is inaccessible and multiple teams finetune the same model in parallel.
>
> In contrast, merging across model families requires tackling architectural and tokenization mismatches, where parameter-level arithmetic is not directly applicable. Prior works have explored combining knowledge from heterogeneous models, such as model routing that dynamically selects among models at inference time [1], or model ensembling that aggregates outputs [2]. While important in their own right, these directions constitute separate problem formulations that are not directly comparable to our work. While extending merging to heterogeneous base models is a promising direction, we believe the within-family merging problem remains a rich and impactful domain with immediate utility.
>
> [1] A Survey on Model MoErging: Recycling and Routing Among Specialized Experts for Collaborative Learning. Yadav et al, 2024.
>
> [2] Determine-Then-Ensemble: Necessity of Top-k Union for Large Language Model Ensembling. Yao et al, 2024.
>
>
> ---
>
> **[Q1 - Caption]** MTL refers to multi-task learning in our context. We used this abbreviation in the main text on line 212, and we will revise the caption as well in our final manuscript.
>
> ---
>
> **[Q2 - Performance]** Our intent was to highlight that Localize-and-Stitch (L&S), RegMean, and Task Arithmetic achieve a favorable trade-off between performance and efficiency *relative to other methods*. We will clarify this in the final manuscript to avoid misinterpretation.

---

### Comment · Area_Chair_73JN · 2025-08-06
**Discussion period has been extended**

Dear reviewers,

the discussion period has been extended. Please read the authors' rebuttal and other reviewers' comments. You are encouraged to provide your further feedback and engage in a discussion with the authors.

Your AC

---

### Note · Authors · 2025-08-12

We thank all reviewers for their thoughtful feedback and the time invested in evaluating our work. We especially appreciate the recognition of several key contributions:
- **Comprehensive and realistic benchmark** using state-of-the-art open-source models and five diverse, real-world task domains, which offers a more relevant and scalable evaluation of model merging (Reviewer dFEo, 9W6o, SK6G, buXu).
- **Systematic and controlled comparisons** of eight merging algorithms using carefully constructed domain-specialized models, all under a unified training/validation protocol (Reviewer dFEo, 9W6o, SK6G, buXu).
- **Holistic evaluation framework** that goes beyond task performance to include forgetting analysis, runtime efficiency, and empirical insights (Reviewer dFEo, 9W6o, SK6G, buXu).
- **Clarity and practical utility**, with informative figures, actionable guidelines for practitioners, and detailed documentation for reproducibility (Reviewer dFEo, SK6G).

During the rebuttal, we worked diligently to address the remaining concerns:
- **Expanded model families**: We added results with Qwen3-4B as the base model, with plans for continued expansion in future updates (Reviewer SK6G, buXu).
- **Clarified scope**: We discussed why merging MoE models and merging different base models pose fundamentally different problems to our setting (Reviewer dFEo, 9W6o, SK6G).
- **Detailed results**: We included finetuning runtime, per-task performance, and additional hyperparameter tuning details (Reviewer 9W6o, buXu).
- **Deeper insights**: We provided more insights on the effectiveness of model merging, and connected our empirical findings to theoretical understanding of multi-task learning and task arithmetic (Reviewer SK6G, 9W6o, buXu).

Reviewers have acknowledged these concerns as addressed, and we will incorporate the additional results, discussions, and feedback into the final manuscript.

---

### Decision · Program_Chairs · 2025-09-18

**Decision:**

Accept (poster)

**Comment:**

This submission introduces MergeBench, a benchmark designed to evaluate model merging techniques for large language models (LLMs) spezalized on domains such as instruction following, mathematics, multilingual understanding, coding, and safety. The benchmark evaluated eight model merging methods according to three dimensions (multi-task performance, forgetting, and runtime efficiency). This paper provides valuable insights about the performance of model merging techniques and a solid foundation for method benchmarking in this domain.

This submission received four thoughtful and detailed reviews with an average rating of 4.5. The authors have provided a detailed rebuttal to address the questions and concerns raised by the reviewers, which was acknowledged by the reviewers.

Based on the reviewers' feedback, the rebuttal and the discussion, and finally, the overall rating, I recommend the paper for **acceptance**. I would like to ask the authors to include all recommended improvements as given by the reviewers for the final version of the paper.

===== FINAL UPDATE FROM DB Track PCs ====

The final decision for this paper has been taken by the program chairs after consultation with the SACs. All Senior Area Chairs have ranked papers according to the feedback from the AC during the review process. We decided to leave the original meta-review to reflect the opinion of the AC in light of the initial discussions with reviewers and SAC.